# Host–Pathogen Interactions of Marine Gram-Positive Bacteria

**DOI:** 10.3390/biology11091316

**Published:** 2022-09-05

**Authors:** Hajarooba Gnanagobal, Javier Santander

**Affiliations:** Marine Microbial Pathogenesis and Vaccinology Laboratory, Department of Ocean Sciences, Faculty of Science, Memorial University of Newfoundland, St. John’s, NL A1C 5S7, Canada

**Keywords:** Gram-positive pathogen, virulence, fish immune response

## Abstract

**Simple Summary:**

Complex interactions between marine Gram-positive pathogens and fish hosts in the marine environment can result in diseases of economically important finfish, which cause economic losses in the aquaculture industry. Understanding how these pathogens interact with the fish host and generate disease will contribute to efficient prophylactic measures and treatments. To our knowledge, there are no systematic reviews on marine Gram-positive pathogens. Therefore, here we reviewed the host–pathogen interactions of marine Gram-positive pathogens from the pathogen-centric and host-centric points of view.

**Abstract:**

Marine Gram-positive bacterial pathogens, including *Renibacterium salmoninarum*, *Mycobacterium marinum*, *Nocardia seriolae*, *Lactococcus garvieae*, and *Streptococcus* spp. cause economic losses in marine fish aquaculture worldwide. Comprehensive information on these pathogens and their dynamic interactions with their respective fish–host systems are critical to developing effective prophylactic measures and treatments. While much is known about bacterial virulence and fish immune response, it is necessary to synthesize the knowledge in terms of host–pathogen interactions as a centerpiece to establish a crucial connection between the intricate details of marine Gram-positive pathogens and their fish hosts. Therefore, this review provides a holistic view and discusses the different stages of the host–pathogen interactions of marine Gram-positive pathogens. Gram-positive pathogens can invade fish tissues, evade the fish defenses, proliferate in the host system, and modulate the fish immune response. Marine Gram-positive pathogens have a unique set of virulence factors that facilitate adhesion (e.g., adhesins, hemagglutination activity, sortase, and capsules), invasion (e.g., toxins, hemolysins/cytolysins, the type VII secretion system, and immune-suppressive proteins), evasion (e.g., free radical quenching, actin-based motility, and the inhibition of phagolysosomal fusion), and proliferation and survival (e.g., heme utilization and siderophore-mediated iron acquisition systems) in the fish host. After infection, the fish host initiates specific innate and adaptive immune responses according to the extracellular or intracellular mechanism of infection. Although efforts have continued to be made in understanding the complex interplay at the host–pathogen interface, integrated omics-based investigations targeting host–pathogen–marine environment interactions hold promise for future research.

## 1. Introduction

Marine Gram-positive bacteria include two major subdivisions, the phylum *Actinobacteria,* with high guanine and cytosine (G + C) contents in their genomes, and the phylum *Firmicutes*, with low (G + C) contents [1]. In most marine environments, Gram-positive bacterial abundance is smaller compared to Gram-negative bacteria [2,3,4], and the presence of Gram-positive bacteria in marine sediments could be linked to nutrient availability [2].

Most marine Gram-positive bacteria have a land origin, and it is believed that they were introduced into marine environments from terrestrial soils [5,6]. For instance, *Arthrobacter* spp., which are soil bacteria, are the closest relatives of *Renibacterium salmoninarum*, a pathogen of marine and freshwater fish [7]. The *R. salmoninarum* genome (~3 Mb) had a significant reduction compared to the *Arthrobacter* spp. genome (~5 Mb) and other Gram-positive environmental bacteria, reflecting its parasitic lifestyle within the host [8].

Non-pathogenic marine Gram-positive bacteria could benefit the host and have practical utilizations as probiotics in aquaculture (e.g., *Lactococcus* spp.) [9,10]. For instance, *Lactococcus lactis* isolated from the gastrointestinal tract of a wild olive flounder (*Paralichthyes olivaceus*) conferred protection against *Streptococcus parauberis* through competitive exclusion [11].

Although Gram-negative bacteria are the most significant pathogens of wild and cultured fish (i.e., *Vibrio* spp., *Aeromonas* spp., and *Edwardsiella* spp.), Gram-positive pathogens, including acid-fast bacteria, can also cause severe economic losses to the marine finfish aquaculture industry, but they are less frequently reported [12,13].

Only a few marine Gram-positive bacteria are primary pathogens (e.g., *M. marinum* and *R. salmoninarum*) [14,15], and the majority are considered opportunistic, causing disease if they are present in high numbers or infect immunocompromised hosts [12,13]. Intracellular marine Gram-positive pathogens (*R. salmoninarum*, *Mycobacterium*, and *Nocardia* spp) can cause chronic persistent infections [16], and several extracellular Gram-positive cocci (e.g., *Lactococcus garviae* and *Streptococcus iniae*) can affect the central nervous systems of fish [17].

Fish diseases continue to be a significant economic threat in aquaculture worldwide and a concern for wild fish populations, especially under the current climate change scenario. Understanding the host-pathogen interactions of marine Gram-positive bacteria will help improve current prophylaxis strategies and the development of novel strategies to prevent infectious diseases in aquaculture environments.

A pathogen is a microbe that can cause disease to the host, and this ability also depends on host immunity. Pathogenicity means the “potential of a microbe to cause damage in a host.” On the other hand, virulence is either the “degree of pathogenicity” or the “relative capacity of a microbe to cause damage in a host” [18]. After a successful infection, the host is damaged due to either direct microbial activity or an uncontrolled host immune response [18], which ultimately affects host homeostasis [19].

The host–pathogen interaction is a trade-off between host and pathogen that depends on the environmental conditions [20,21]. Methot and Alison (2014) suggest that virulence is an outcome of a specific host-pathogen interaction, not a fixed microbial or host property [21] (Figure 1A). For instance, a virulent microbe can become avirulent or less pathogenic in an immune host, whereas an avirulent microbe can become virulent (i.e., pathogenic) in an immunocompromised host [22]. Infectious diseases occur when a susceptible host and a virulent microbe meet in an environmental context that facilitates such an occurrence (i.e., environmental stressors in the marine environment, high stocking densities in cultured conditions, and parasitic infestations) [23,24] (Figure 1B). The trade-off between host and pathogen could result in fitness-related costs to both the host (i.e., measurable damage) [13] and the pathogen (i.e., limited ability to spread within the host) [21] (Figure 2).

Climate change is currently affecting several food-producing sectors, and marine aquaculture has already been impacted. Extreme high temperatures in summers and extremely low temperatures in winter lead to immune suppression of farmed fish, increasing the susceptibility to infectious diseases. Effects of climate change on the virulence and evolution of marine Gram-positive bacterial pathogens have not been addressed. Moreover, how the current environmental and ecological changes (e.g., the migration of invasive species) affect the host–pathogen interactions and how changes in this interplay affect therapeutic and prophylactic measurements have yet to be investigated. Considerable attention has been devoted to studying marine Gram-positive pathogenesis and fish immune responses. However, research using multidisciplinary analyses (i.e., integrated omics) to study the host-pathogen-environment interactions of marine Gram-positive bacteria is insufficient and requires future investigation.

This review provides a comprehensive synopsis of how economically important marine Gram-positive bacterial pathogens (Table 1) adhere, invade, evade, proliferate, and cause damage in the fish host (i.e., pathogen-centric approaches) and how the host responds and controls the invader (i.e., host-centric approaches).

## 2. Pathogen-Centric Approaches

### 2.1. Adhesion/Host Recognition

Pathogen adherence to host surfaces (cells or substrates) is the first step that initiates the host-pathogen interaction, and it is a prerequisite for invasion [25]. One of the factors affecting bacterial adhesion to the host surface (e.g., fish mucus) is bacterial hydrophobicity [26]. High bacterial hydrophobicity is correlated with high adhesion, and therefore this phenotype has an important role in pathogenicity [27]. For instance, the higher surface hydrophobicity and hemagglutinating activity of *Streptococcus dysgalactiae* correlate with its strong adherence ability *in vitro* to carp epithelioma papillosum cells (EPCs). The relationship between hydrophobicity and virulence has also been reported in *R. salmoninarum*, where virulent strains with hydrophobic cell surfaces showed higher adherence and auto-agglutination [28].

Bacterial adhesins made up of proteins and carbohydrates, enable interaction with the adhesive molecules on the host tissue surface. Protein adhesins include fimbrial (or pili) and afimbrial structures [25]. Bacterial adhesins have been identified based on the bacterial hemagglutination potential [29]. Even though *S. dysgalactiae* and *L. garviae* have fimbria-like structures on their surfaces, *S. dysgalactiae* isolates showed hemagglutination, while *L. garviae* did not, suggesting that the fimbria-like structures of *S. dysgalactiae* were functionally mediating its hemagglutination activity [27,30,31]. The role of surface-anchoring M family proteins as adhesins is well-known [32,33]. For example, the *S. iniae simA* gene encodes an M-like protein that contributes to adhesion, subsequent invasion, and phagocytic killing resistance (Figure 3). An *S. iniae simA* mutant provided 100% protection in hybrid striped bass (*Morone chrysops* × *Morone saxatilis*), and it could be utilized as an effective live attenuated vaccine [34].

The major soluble antigen (*msa*) p57, which is both a major outer membrane (70% of the surface protein) and a secretory protein, is the main virulence factor of *R. salmoninarum* [7]. p57 binds to eucaryotic cells and causes immune suppression [7,35]. Because of its hydrophobic and hemagglutinating characteristics, the p57 monomer resembles bacterial adherence structures (i.e., fimbrial adhesins) and could facilitate adhesion to host cells [36,37,38]. 

**Table 1 biology-11-01316-t001:** Host–pathogen interactions of significant marine Gram-positive bacteria.

Pathogen	Disease (Water Temperature)	Host(s) Marine Fish ^1^	Damage to the Host ^2^	Main Virulence Factors	References
Aerobic acid-fast rods and cocci					
*Mycobacterium* spp. *M. chelonei subsp. piscarium* *M. fortuitum* *M. marinum* *M. neoaurum*	Mycobacteriosis/fish tuberculosis (17–30 °C)	Most fish spp: turbot (*Scophthalmus maximus*), Atlantic salmon (*Salmo salar*), chinook salmon (*Oncorhynchus tshawytscha*), coho salmon (*Oncorhynchus kisutch*), sea bass (*Lateolabrax japonicus*)	a. Scale loss, dermal ulceration, pigmentary changes, abnormal behavior and emaciation, and ascites; b. Hemorrhagic ascites, nodular lesions in spleen, liver, kidney; c. Granulomatous inflammation	SecA2 substrate-PknG, PE, PPE family proteins; T7SS, mycolactone, *iipA* gene - invasion and intracellular persistence protein	[39,40,41]
*Nocardia* spp. *N. asteroides* *N. salmonicida* *N. seriolae*	Nocardiosis (24–28 °C)	Most fish spp: grey mullet (*Mugil cephalus*), seabass, largemouth bass (*Micropterus salmoides*), yellowtail (*Seriola quinqueradita*)	a. Erratic swimming, anorexia; b. White-yellow nodules in spleen, kidney, and liver; c. Granulomatous lesions with necrosis	ATP-binding cassette transporters, capsule, sortase A, ESX-1, fibronectin-binding protein, myosin cross-reactive antigen, serine protease, virulence genes for cell invasion and alteration of phagocytic function	[42,43]
Aerobic rods and cocci					
*Renibacterium salmoninarum*	Bacterial Kidney Disease (8–15 °C)	1. Salmonids: Atlantic salmon, brown trout (*Salmo trutta*), rainbow trout (*Oncorhynchus mykiss*), chinook salmon, coho salmon; 2. non-salmonids: ayu (*Plecoglossus altivelis*), north Pacific hake (*Merluccius productus*), Pacific herring (*Clupea pallasii pallasii*), sablefish (*Anoplopoma fimbria*)	a. Skin darkening, lethargy, ascites, exophthalmia, skin blisters, hemorrhages around the vent, shallow skin ulcers, large cystic cavities in the skeletal muscle; b. Greyish-white nodular lesions in kidney, spleen, liver; enlarged spleen and kidney, pseudomembrane in internal organs, turbid fluid in abdominal/pericardial cavities; c. Bacteremia with chronic granulomatous inflammation	Hemolytic, proteolytic, catalase, DNase, and iron reductase activities, exotoxin, virulence genes - hemolysin (*rsh*), a zinc-metalloprotease (*hly*), glucose kinase; capsule, fimbriae, immunosuppressive proteins p57 and p22	[7,44]
*Rhodococcus* sp.	Ocular edema(12 °C)	Atlantic salmon, chinook salmon	a. Ocular melanosis; b. Ocular lesions, nodules in muscle and organs; c. Granulomas in kidney	Very low-level mortality with high dose (5 × 10^8^ bacteria/fish)	[45]
The “lactic acid bacteria”.					
*Lactococcus garviae*	Lactococcosis (16–18 °C)	Most fish spp: yellowtail, grey mullet, Japanese or olive flounder (*Paralichthys olivaceus*), rainbow trout	a. Exophthalmia, lethargy, erosion of tail fin, redness of anal fin, petechiae inside operculum; b. Hemorrhages and petechias at the internal organs’ surface; c. Ocular lesions have fibrous tissue formation with infiltrated inflammatory cells	Hemolysins, capsule, cell-associated toxin NADH oxidase, superoxide dismutase, adhesins, sortase, and phosphoglucomutase encoding genes	[30,46,47]
*Streptococcus iniae*	Streptococcosis/Meningoencephalitis (15–18 °C)	Most fish spp: yellowtail, olive flounder, sea bass, barramundi (*Lates calcarifer*), European seabass (*Dicentrarchus* *labrax*), gilthead seabream (*Sparus aurata*)	a. Exophthalmia, petechiae around mouth, anus, fins, loss of orientation exophthalmia; b. Fluid in peritoneal cavity; c. Intravascular lesions leading to pericarditis, focal necrosis in liver, spleen, and kidney	Capsular polysaccharide, phosphoglucomutase, fibronectin binding proteins, streptolysin, hemolysins, plasminogen-binding protein, *simA* M-like protein	[34,48,49,50]
*Streptococcus parauberis*	Streptococcosis (>15 °C)	Turbot	a. Bilateral exophthalmia, emaciation; b. Hemorrhages in anal and pectoral fins and eyes, pale liver, congested kidney and spleen; c. Hemorrhagic inflammation in intestine	*simA* encoding M-like protein, *hasA* and *hasB* genes for capsule production and phagocytic resistance	[51,52]
*Streptococcus dysgalactiae*	Streptococcosis (>15 °C)	Amberjack (*Seriola dumerili*), yellowtail	a. Typical form of necrosis in the caudal peduncle; b. Septicemia	Cell hydrophobicity, M protein, streptolysin S, super antigen, streptococcal pyrogenic exotoxin G	[27,31,53]
*Streptococcus phocae*	Streptococcosis (5–15 °C)	Atlantic salmon	a. Exophthalmia, hemorrhagic eyes with accumulation of purulent fluid, skin abscesses; b. Hemorrhage in the abdominal fat, pericarditis and enlarged liver, spleen, and kidney; **c**. Pathological lesions in the spleen, liver, heart, and muscle, leucocytic perivascular infiltration in spleen, moderate vascular degeneration in the liver.	Hemolysins, collagen adhesion protein, capsule, cell hydrophobicity	[54,55]

^1^ Marine fish hosts for the respective marine Gram-positive bacteria were gathered by considering Austin and Austin (2016) [13]. ^2^ Damage from direct bacterial damage and host pathology: (a) external signs; (b) internal signs; (c) histopathology [12,56].

**Figure 3 biology-11-01316-f003:**
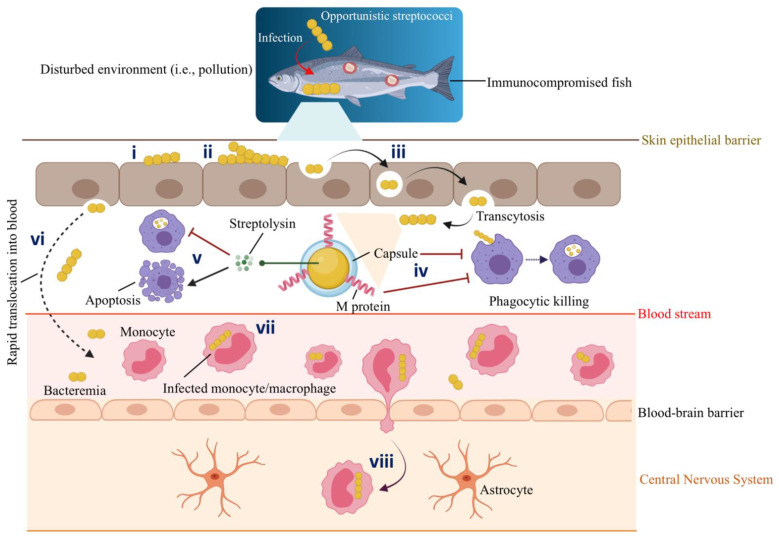
Schematic representation of host-pathogen interactions between marine fish and opportunistic *Streptococcus* spp. Gram-positive streptococci infect immunocompromised fish with a decreased immune response that lives in a conducive environment (e.g., polluted marine environment) that facilitates such an infection: (**i**) Adhesion. (**ii**) Colonization of the epithelial barrier (i.e., mucus). (**iii**) Epithelial invasion by transcytosis. (**iv**) Antiphagocytic factors such as capsule and M protein aid *S. iniae* to survive phagocytic killing. (**v**) Streptolysins secreted by *S. iniae* inhibit phagocytic killing and, at the same time, induce the apoptosis of phagocytes. Thus, infected macrophages undergo apoptotic death and fail to prime a specific immune response. (**vi**) Invasive streptococci persist intracellularly for a short time and rapidly translocate into the blood circulation system. (**vii**) *S. iniae* hijack the migrating monocytes or macrophages. For instance, Zlotkin et al. (2003) [57] observed the presence of 70% of the *S. iniae* in the infected monocytes in the blood of diseased fish. (**viii**) Infected monocytes/macrophages act as trojan horses that carry *S. iniae*, cross the blood-brain barrier, transmigrate, and deliver bacteria to the fish central nervous system (CNS). Thus, *S. iniae* can enter the CNS through its association with the migrating monocytes. This original illustration was generated by authors using BioRender (https://biorender.com/) (accessed on 3 March 2022).

For example, the peritrichous fimbriae of *R. salmoninarum* have been shown to be composed of p57 [38]. The biological functions of p57, such as binding and agglutinating fish leucocytes, enable *R. salmoninarum* adhesion and invasion [36].

Intriguingly, a recent proteome study found a high abundance of p57, p22 (a second key immunosuppressive protein), and proteins implicated in bacterial adhesion in membrane vesicles of *R. salmoninarum*, suggesting that the membrane vesicles could play a role in the pathogen attachment and subsequent BKD development [58,59,60].

Purified p57 loses its immunosuppressive activity when treated with a temperature-dependent endogenous *R. salmoninarum* serine protease [61]. This protease could post-translationally modulate the function of p57 by altering the amount of functionally active p57 at the bacterial surface [7]. Moreover, iron-limited conditions reduced p57 processing into mature and functional protein [62] and, in contrast, facilitated the overproduction of p57, according to a proteomic analysis [63]. Thus, the iron-restricted conditions of fish serum, as the means of nutritional immunity during the early infection stages of *R. salmoninarum,* might affect p57 stability or expression before the intracellular invasion [7]. Overall, the hydrophobic surface protein p57 binds with the fish host cell receptors during adhesion (Figure 4) and contributes to the pathogen’s entry [36].

Fish mucosal surfaces, besides being a physical barrier to pathogens, have antibacterial molecules (e.g., immunoglobulins, antimicrobial peptides, etc.) to prevent infections. However, pathogens have developed mechanisms to overcome this immune defense [64]. For example, the virulence of *S. phocae* is attributed to its capsule, which allows the pathogen to adhere to the Atlantic salmon (*Salmo salar*) mucus and withstand the mucus and serum’s bactericidal activity [65] (Figure 3).

Another virulence factor related to adhesion is sortases, which have a role in covalent anchoring cell surface proteins in Gram-positive bacteria and contribute to bacterial virulence and modulation of the host immune system. The importance of sortase in the adhesion/invasion and virulence of *R. salmoninarum* was demonstrated by Sudheesh et al. (2007) [66]. Reduced virulence in *R. salmoninarum* was observed after treating the pathogen with phenyl vinyl sulfone (PVS), a sortase inhibitor. PVS-treated bacteria showed reduced binding to chinook salmon (*O. tshawytscha*) fibronectin, a ligand for many bacterial adhesins that is abundantly present in eukaryotic extracellular matrix and plasma [67]. Moreover, it showed inhibited cell adherence, invasion, replication, and cytopathic effects on chinook salmon embryo cells compared to the untreated bacteria. In addition, the inhibition of sortase activity has potential use in anti-virulence chemotherapy [68].

Overall, studies focusing on anti-adhesion therapies, the use of potential drugs that block adhesion (i.e., PVS), and the design of DNA vaccine encoding adhesins will contribute to preventing Gram-positive infections in marine farmed fish.

### 2.2. Invasion

Invasion is the ability of a pathogen to spread to different tissues or organs and/or enter host cells. Once the pathogen adheres and colonizes at the mucosal host surfaces, it obtains deeper access into the host, allowing it to sustain the infection cycle [25]. Gram-positives can extracellularly invade by breaking down the tissue boundaries and dispersing in the host while remaining outside of host cells or can intracellularly invade and persist within the host cells [25,69].

Streptococci are usually extracellular pathogens, but several strains are capable of invading eucaryotic cells [70]. Some *S. phocae* isolates attached to the chinook salmon embryo (CHSE) cell line (adhesion values: 18.7–145.3%), but they were unable to intracellularly invade (invasion values: 0–0.42%), suggesting a lack of structural components/pathways to facilitate the intracellular invasion [65]. In contrast, some *S. phocae* isolates reach the cytoplasm of CHSE cells at 2 and 20 h post-infection, implying that *S. phocae* is using its virulence mechanism to find a nutritionally compatible niche (i.e., the cytosol) within the host to support its further proliferation and survival [71]. Moreover, Eyngor et al. (2007) demonstrated the critical role of the intracellular epithelial invasion of *S. iniae* for rapid translocation to internal tissues and further infection in rainbow trout (*O. mykiss*) [72] (Figure 3). Overall, pathogens employ virulence mechanisms to navigate through the extracellular matrix, breach the barriers between tissues, extend into adjacent tissues or cells and obtain factors (i.e., nutrients) that sustain their growth.

Gram-positive marine pathogens also synthesize the toxins required for intracellular invasion. *R. salmoninarum* secreted an unknown exotoxin that is lethal to Atlantic salmon fingerlings (9–12 g) at an intraperitoneal (i.p.) dose of 160 μg [73]. Mycolactone F, a *Mycobacterium* spp. toxin that causes apoptosis and necrosis, has been purified from the fish pathogens *M. marinum* and *M. pseudoshottsii* [74]. However, the role of this toxin has not been characterized in fish cells [40].

Extracellular toxins, such as hemolysins and cytotoxins, are essential for systemic infection during the extracellular invasion. These toxins lyse erythrocytes and release iron, heme, or hemoglobin for bacterial growth by forming pores on them or altering phospholipid structures in the membrane [25,75,76]. Hemolysins and genes related to hemolytic activity have been reported in marine Gram-positive bacteria. For instance, *S. iniae* secretes a β-hemolytic streptolysin S (SLS) homolog, a pore-forming cytotoxin [77]. A loss of SLS production in *S. iniae* caused virulence attenuation in a hybrid striped bass host. SLS contributes to *S. iniae* virulence by causing local tissue necrosis, helping the pathogen to resist phagocytic killing (Figure 3) [77]. The genome of *R. salmoninarum* contains three hemolysin encoding genes [8], which could be critical for its intracellular infection and progression. One of these *R. salmoninarum* hemolysins was recently described as helping *R. salmoninarum* cope with stressful conditions in the host during iron limitation [63]. Moreover, the expression of the hemolysin genes *hly1* and *hly2* in the α-hemolytic bacterium *L. garviae* is associated with its pathogenicity [47]. Another protein related to extracellular invasion is α-enolase. This cell-wall-associated and plasminogen-binding protein of *S. iniae* is partially responsible for tissue invasion. *S. iniae* crossed tissue barriers through plasminogen activation and might migrate faster in the fish extracellular matrix using the proteolytic activity of plasmin [78,79].

Bacterial secretion systems aid pathogenic bacteria in secreting virulence factors (i.e., effector proteins) from bacterial cytosol into host cells during the intracellular invasion, and they can target professional and non-professional phagocytic cells [25,80]. There are two main invasion mechanisms that facilitate bacterial internalization into host cells, called trigger and zipper. In the trigger mechanism, bacteria transfect effectors into the cytoplasm of the host cell via specific secretion systems, causing massive cytoskeletal rearrangements and the development of ruffles, allowing the bacterium to be internalized [81]. For instance, *Mycobacterium* spp. use the type VII secretion system (T7SS), which is encoded by the *esx* loci (*esx1*–*5*), for intracellular protein trafficking and macrophage survival [82,83,84]. *M. marinum esx1* is essential for infection in the fish host [85]. *esxA* and *esxB* are essential effectors translocated by the ESX-1 system [85,86]. *esx5* has been identified as an active protein secretion system in *M. marinum*, translocating a variety of PE (Pro-Glu protein) and PPE (Pro-Pro-Glu protein) effector proteins [84,87]. Several of these proteins are found on the surface of mycobacterial cells, which explains their interactions with host cells [88,89,90]. *M. marinum esx5* mutants were utilized by Abdallah et al. (2008) to demonstrate the role of *M. marinum* ESX-5 in triggering host cell death and modulating macrophage cytokine responses [91]. *M. marinum esx5* mutants were slightly attenuated in the zebrafish embryos but were hypervirulent in the adult zebrafish, which was characterized by higher *esx5* mutant bacterial loads and the early initiation of granuloma formation [92]. This difference in virulence between the embryo and the adult zebrafish does not appear to be mediated by the adaptive immune system since the *rag*-deficient zebrafish, which lack functional B and T lymphocytes, also exhibited the hypervirulence phenotype. Therefore, other factors that differ between embryonic and adult zebrafish may mediate *M. marinum* hypervirulence in adult zebrafish. For instance, it could be caused by more local and possibly intracellular effects that result from the interplay between the fish host and *M. marinum* rather than a general immune response or modified extracellular environment [92]. A new subclass of the type IV secretion system (T4SS), type-IV-C was proposed in the Gram-positive genus *Streptococcus* in humans, which could mediate DNA transfer across the cell envelope and enhance bacterial pathogenicity [93]. Interestingly, proteins from T4SS have been identified as virulence factors in *R. salmoninarum* [63]. The presence of this novel secretion system in marine Gram-positive streptococci, however, has yet to be reported and opens avenues for future research.

In the zipper mechanism, the interaction of bacterial surface proteins with host proteins causes cytoskeleton and membrane rearrangements, resulting in the pathogen’s internalization [81]. Because *R. salmoninarum* has an affinity for phagocytes, sinusoidal cells, and reticular and barrier cells, a putative mechanism (i.e., zipper) (A in Figure 4) of its intracellular invasion has been linked to the surface protein p57 [94]. This mechanism involves C3b, a complement pathway opsonin, binding to the bacterial surface, ligation to C3b-receptor-bearing salmonid phagocytes, and subsequently increased internalization [95] (Figure 4). In a histopathological examination, Bruno (1986) detected live *R. salmoninarum* cells in phagocytes of the kidneys and spleens of rainbow trout and Atlantic salmon 45 min after i.p. injection and high numbers of bacteria in macrophages after 6–10 days [96].

Overall, most known invasion mechanisms of marine Gram-positive bacteria proceed via protein-protein interactions. Studies focusing on reducing cellular invasiveness and weakening the interactions between pathogen surface proteins and fish host proteins in the extracellular matrix would be beneficial in giving insights into chemotherapeutic treatments in aquaculture.

### 2.3. Evasion

Bacterial immune evasion is a process by which pathogens avoid or inactivate the host immune response once they gain access to the intracellular host milieu. Waxy hydrophobic cell walls or mycolic acids in the mycobacterial capsule prevent digestion by lysosomal enzymes during evasion [97]. For instance, capsules in *L. garviae* and *S. iniae* enable these pathogens to resist phagocytosis by macrophages [50,98].

Various bacterial pathogens have adapted to survive and multiply within host cells (i.e., professional phagocytes and non-phagocytic cells) after the invasion. The interaction of *S. iniae* with fish phagocytes (Figure 3) is crucial in its evasion and contributes to its virulence [99]. Fish infected with *S. iniae* showed evident bacteremia, and diseased fish hold up to 70% of the bacteria in the blood within its phagocytes [57]. After the invasion, *S. iniae* survived phagocytic killing and rapidly disseminated to systemic tissues through the blood. According to Zlotkin et al. (2003), *S. iniae* hijacked the peripherical monocytes/macrophages and used them as trojan horses to enter the central nervous system [57]. Moreover, this pathogen effectively circumvented the host’s immune system by inducing apoptosis in fish macrophages [57].

Since professional phagocytic cells, such as macrophages or neutrophils, have mechanisms to eliminate ingested bacteria, the survival and replication inside them are remarkable. One of these killing mechanisms is the production of reactive oxygen species (ROS). The pathogen’s ability to resist the host oxidative burst caused by the ROS (i.e., superoxide (O_2_^−^) and hydrogen peroxide (H_2_O_2_)), which are produced in phagocytic vacuoles [100,101], is related to free radical quenching (Figure 4). The microbicidal component of the phagosome, NADPH oxidase, generates ROS [102]; the superoxide dismutase (SOD) converts O_2_^−^ to H_2_O_2_, and the bacterial catalase converts H_2_O_2_ to O_2_ + H_2_O to prevent damage. The higher O_2_^−^ production by rainbow trout macrophages in response to heat-killed opsonized *R. salmoninarum*, in contrast, to live or UV-killed bacteria, showed that the bacterium’s catalase and SOD quench the macrophages’ O_2_^−^ production [103,104,105]. Moreover, the *R. salmoninarum* genome contains genes for thioredoxin peroxidase and the SOD enzymes that confer resistance to oxygen radicals [8].

After the intracellular invasion, pathogens can reside in three intracellular niches: the phagolysosome, phagosome, and host cell cytosol. Another host mechanism to eliminate bacteria is lowering the pH of pathogen-containing vesicles. To bypass this antibacterial mechanism, bacterial pathogens can survive and multiply in the phagolysosome (pH = 5.0–5.5), preclude the formation of the phagolysosome, or escape to the host cell cytoplasm [101,102]. After *R. salmoninarum* is phagocytized, this bacterium escaped to the host cytosol by disrupting or lysing the phagosome membrane (Figure 4) [106]. *R. salmoninarum* hemolytic proteins, p57 antigen, hemolysin (*hly*), and cytolysins (*rsh*) facilitated the budding out from the phagosome to the host cell cytoplasm [62,106,107,108,109].

The intracellular survival and replication of *R. salmoninarum* are attributed to its cell wall resistance to lysozyme and slow growth rate [15]. While optimal growth rates are required to initiate infection in the host, *R. salmoninarum* switches to suboptimal growth rates (e.g., slow growth) to maintain its intracellular survival [100]. For example, this switching has been observed within the macrophages infected *in vitro* with *R. salmoninarum*, where the pathogen showed a decreased growth rate during the chronic infection [106]. Here, a slower growth rate makes the *R. salmoninarum* dormant, thereby resisting the action of antibiotics targeting actively replicating bacteria. Overall, *R. salmoninarum* exhibits prolonged persistence (i.e., chronic intracellular survival) along with a decreased growth rate in the host-pathogen trade-off (Figure 2B) [20]. Otherwise, rapid intracellular growth kills the host cells, which is a disadvantage for the long-term survival of the pathogen in fish. Dormant *R. salmoninarum* can then make its own “wake-up call” under favorable conditions (i.e., when fish are under stress) and start optimal replication through resuscitation-promoting factors [8,110,111].

Bacteria that survive intracellularly either multiply and spread to cells in the infected tissues or migrate to adjacent tissues from the primary site of colonization. *M. marinum* uses two methods to evade phagocytosis, it escapes from phagosomes (Figure 5A) and/or blocks phagolysosome fusion (Figure 5B). *M. marinum* can escape from the phagosome to the cytosol, where it can recruit host cell cytoskeletal factors to induce actin-based motility that leads to direct cell-to-cell spread [112]. The late phagosome fuses with a phagocytic lysosome during phagosome maturation into a phagolysosome, which is induced by the vesicle-mediated delivery of antimicrobial effectors such as proteases, antimicrobial peptides, and lysozyme [113]. Evidence for the phagolysosomal fusion of *M. marinum* phagosomes was observed in striped bass (*Morone saxatilis*) peritoneal macrophages and rainbow trout primary macrophages [114,115]. The morphological presence of the intact mycobacteria within phagolysosomes indicated the pathogen’s ability to withstand the hostile phagolysosomal environment [114,115]. In contrast, pathogenic *M. marinum* inhibited phagosome–lysosome fusion in fish monocytes and resided within unfused vacuoles of the carp leucocyte culture cells infected with *M. marinum*, which did not acidify [116]. Mycobacterial protein kinase G (Pkng), secreted by SecA2 into the cytosol of infected host cells, is implicated in preventing phagosome-lysosome fusion and facilitating the intracellular survival of mycobacteria [117,118,119,120]. *M. marinum* intracellular survival and virulence have been linked to the Pkng and SecA2 pathway [121,122]. For example, mycobacteria were incapable of preventing phagosomal maturation when SecA2 mutated [121]. On the other hand, the restoration of a phagosomal maturation block by overexpressing the PknG in SecA2 mutants suggests a role of PknG in mycobacterial pathogenicity [122].

Investigating the insights of bacterial mechanisms related to the survival of mycobacteria within phagolysosomes will be noteworthy. For instance, Parikka et al. (2012) demonstrated the latency mechanism of *M. marinum* in a zebrafish model [124]. Mycobacteria became dormant in response to the immune response to an infection and hypoxia and were reactivated by an *ex vivo* resuscitation-promoting factor addition [124]. Thus, research on host-pathogen-environment interactions is essential to predict or avoid the risk of reversion of latent mycobacterial infection in wild or cultured fish populations from polluted marine environments or sea farms.

### 2.4. Proliferation and Survival Inside the Host

After entry, pathogens commandeer and use the host cells, not only for their own replication and survival but also for thriving in the host [125]. Pathogens, particularly opportunistic pathogens, proliferate more easily within the host than they do outside the host [126]. *R. salmoninarum* cannot survive for extended periods of time outside of its host, and the survival times for *R. salmoninarum* in environmental samples ranged from 4 to 21 days at 10–18 °C, which is relatively a short time and suggests that this pathogen replicates within the host rather than in the environment [8,111]. The pathogen requires a minimal set of metabolic pathways and a significantly smaller number of genes to multiply inside the host than in the environment, implying that it obtains a considerable amount of essential nutrients directly from the host and that several of its biosynthetic pathways are inactive when it is inside the host [100]. For instance, the presence of several pseudogenes in the *R. salmoninarum* genome may have contributed to the apparent decrease in many anabolic pathways. Simultaneously, the number of bacterial proteins involved in energy metabolism, transcription, and signal transduction in the genome of *R. salmoninarum* was lower than that of its environmental relative, *Arthrobacter* spp. [8]. The reduction in metabolic pathways and bacterial proteins suggests that *R. salmoninarum* depends on the host for its unique requirements. As a result, *R. salmoninarum* should have evolved to exploit the fish host’s intrinsic machinery.

Nutritional immunity is a mechanism used by the host to restrict the availability of essential nutrients in their tissues and fluids, such as iron and vitamins, and prevent the proliferation of potential pathogenic invaders [75,127,128]. For instance, the battle over limited iron is critical for the host and the pathogen during infections [129]. Iron is a co-factor of many enzymes, and it is involved in bacterial physiological processes, including central metabolism, transcription, and DNA replication [130]. The ability to sequester iron from the host during infection is essential for bacterial virulence and survival [131,132]. This is also critical for marine pathogens outside their hosts since iron richness in marine environments is extremely low (picograms per liter) [17]. Pathogenic bacteria usually have various iron-acquisition mechanisms for ‘iron-piracy’ to circumvent the nutritional immunity within the host [128,133]. Indeed, iron-depleted conditions inside the host act as a signal for the expression of virulence genes [75,134].

Three iron-acquisition mechanisms have been reported in *R. salmoninarum*, including NADPH reductase, siderophore production, and heme utilization [135,136]. Under iron limitation, ferric iron is converted to a ferrous complex by iron reductase and is readily bound to and transported by bacteria [137,138]. Siderophore trafficking in Gram-positive pathogens, which only have a single membrane, is comparatively a simple uptake mechanism compared to Gram-negative bacteria. This mechanism involves a siderophore-binding protein and an associated permease located on the cell membrane [129]. Following iron capture, siderophores bound to receptors on the bacterial surface are internalized, and iron is released in the cytoplasm for growth and colonization during infection. A significant role of iron-acquisition mechanisms (siderophores) in virulence is supported by Bethke et al. (2019) [139,140]. In this study, an *R. salmoninarum* strain (H-2) with a high siderophore production capability was grown under the iron-limited condition and showed significant over-expression of the iron-acquisition-related genes compared to the bacteria grown under normal conditions. On the other hand, *R. salmoninarum* H-2 displayed higher virulence in terms of cytotoxicity, cytopathic effects and induced the expression of pro-inflammatory cytokines in Atlantic salmon kidney cell lines than a strain with lower siderophore production capacity. A proteome analysis of *R. salmoninarum* H-2 grown under iron-limited conditions indicated that the iron homeostasis pathway and critical virulence factors related to iron deprivation were significantly enriched [63].

Genomic analyses of *R. salmoninarum* conducted by Wiens et al. (2008) and Bethke et al. (2016; 2018) revealed important facts about *R. salmoninarum* iron homeostasis [8,136,141]. *R. salmoninarum* has gene clusters that encode for a ferric siderophore import system [8]. According to Bethke et al. (2016), the heme acquisition mechanism of *R. salmoninarum* could be similar to the other Gram-positive bacteria based on the genes that encoded for heme uptake in *R. salmoninarum* (i.e., receptors, permeases, ATPase subunits, and heme oxygenases similar to the HmuTUV (HmuO) ABC transporter system) [136,142,143]. According to Wiens et al. (2008), the heme acquisition operons in the *R. salmoninarum* genome were acquired via horizontal gene transfer during species divergence [8]. Moreover, the increased bacterial resistance to iron toxicity in *R. salmoninarum* grown under iron-limited conditions suggests protection from oxidative stress during intracellular survival [136]. As *R. salmoninarum* falls under high G + C content (56.3%) Gram-positive bacteria, the presence of an additional iron-dependent repressor belonging to the DtxR/IdeR family and their binding sites upstream of important iron-acquisition-related genes were observed [8,141,144,145].

The *S. phocae* isolate of Atlantic salmon secretes siderophores and can acquire heme directly through binding receptors [146]. Interestingly, *S. phocae* expressed an unknown iron-regulated protein (95 kDa) under iron-limited conditions, which could be a receptor for siderophore–iron complexes/heme groups or interact with host-iron-carrying components (e.g., transferrin) [146]. In addition, biofilm formation was observed in the iron-limited condition, indicating *S. phocae*’s ability to sense iron availability in the host. Therefore, the bacterium could develop strategies for bacterial adherence, which leads to successful colonization within the host.

More is known about Gram-negative than Gram-positive iron acquisition systems [147]. Therefore, it is expected that the iron-regulated proteins of marine Gram-positive bacteria require more research. Recent proteomic data obtained from *R. salmoninarum* grown under iron-limited conditions identified important virulence factors related to their iron acquisition mechanisms (e.g., heme uptake and siderophore synthesis), which could aid in designing therapeutic approaches targeting these essential bacterial proteins [63]. For instance, blocking siderophore-mediated iron uptake (e.g., siderophore receptor protein, which is responsible for transporting siderophore–iron complexes into the bacterial cytosol) in Gram-positive pathogens would be an option.

## 3. Host-Centric Approaches—Fish Host Immune Response

Marine Gram-positive bacterial pathogens use diverse mechanisms to infect and manipulate fish host cells and evade immune responses. In contrast, the fish host will mount an immune defense to control the infection and subsequently eliminate it from the system to maintain its homeostasis (Figure 2A). Fish immune responses to marine Gram-positive pathogens have been studied in several fish species (Table 2). Fish immune responses at different stages of infection with marine Gram-positive pathogens are discussed in this section, including innate immunity and pathogen recognition, nutritional immunity, and adaptive immunity.

### 3.1. Toll-like Receptors (Pathogen Recognition)

In fish, TLRs (Toll-like receptors), NLRs (NOD-like receptors), CLRs (C-type lectin receptors), and PGRP (peptidoglycan recognition proteins) are the four main types of pattern recognition receptors (PRRs) [148]. Only TLRs are the subject of this section because they are well-described signalling PRRs in fish innate immunity, detecting Gram-positive pathogen-associated molecular patterns (PAMPs) [149]. Research on the involvement of the other PRRs in Gram-positive bacterial recognition in marine fish has yet to be reported.

TLRs are the innate immune receptors that recognize conserved pathogen molecules (e.g., lipopolysaccharide (LPS), flagellin, and the cell wall) [150] and thereby trigger rapid inflammation and prime adaptive immunity [151,152,153]. The involvement of diverse TLRs in marine fish immunity upon infection with the marine Gram-positive bacteria *L. garviae* [154], *N. seriolae* [155], and *S. dysgalactiae* [156] were reported in transcriptome analyses. In mammals, TLR2 forms a heterodimer with TLR1, which recognizes lipoteichoic acid and peptidoglycan from Gram-positive bacteria [157]. The upregulation of *tlr2* was observed in grey mullet (*M. cephalus*) in response to *L. garviae* [154]. Concurrently, *tlr1* and *tlr2* were upregulated in zebrafish following *M. marinum* infection, which agreed with the known functions of mammalian TLR1 and TLR2 in sensing acid-fast/Gram-positive cell wall components [158]. Though the specific ligand for TLR1 is unknown in fish, *tlr1* showed a similar expression pattern as *ifnγ* (interferon gamma) (i.e., significant upregulation at 6 and 12 hpi) in the Atlantic salmon kidney cell line in response to *R. salmoninarum* [140,150]. This observation is in line with Miettinen et al. (2001), who described the ability of IFN-γ to upregulate TLR1 and TLR2 [159].

Fish TLR5 recognizes bacterial flagellin [160]. However, in response to non-motile (i.e., non-flagellated) Gram-positive pathogens such as *S. iniae* and *R. salmoninarum*, *tlr5* was upregulated in turbot (*S. maximus*) and Atlantic salmon, respectively [161,162,163]. This controversial observation demands future research to study the role of TLR5 beyond the recognition of flagellin.

Mammalian TLR4 recognizes Gram-negative LPS [160]. Teleost fish do not have a complete functional TLR4 [164]. While the presence of TLR4 and some co-receptors was reported in some fish species, the lack of essential co-receptors in teleosts (e.g., CD14) makes this TLR4 not functional for LPS detection in all fish species [157]. Interestingly, the *tlr4* encoding gene in soiny mullet (*Liza haematocheila*) was upregulated in spleens upon Gram-positive *S. dysgalactiae* infection, suggesting an alternative role of TLR4 in the fish immune response to Gram-positive bacteria [156].

Among six non-mammalian fish-specific TLRs (TLR14, 19, 20, 21, 22, and 23) [150], TLR14, TLR20, and TLR 22 showed interactions with marine Gram-positive pathogens. For instance, *S. iniae* infection increased *tlr14* expression in Japanese flounder kidney at 1 dpi [165]. Moreover, upregulated expression levels of *tlr20* and *tlr22* in zebrafish infected with *M. marinum* suggest a role of fish-specific TLR clusters in recognizing bacterial infections [158].

Although the functions of fish-specific TLRs have yet to be reported [150], TLR interactions with Gram-positive fish pathogens suggest that diverse fish TLRs are involved in fish immunity against marine Gram-positive infections.

### 3.2. Nutritional Immunity

The host-mediated withholding of essential nutrients to limit bacterial colonization or nutritional immunity is one of the first lines of defense against bacterial infection. The most significant form of nutritional immunity is iron sequestration in host proteins because iron is essential for bacterial proliferation and virulence [75]. Like in mammals, several host iron-sequestering proteins have been described in marine teleosts, including transferrin [166], ferritin [167], hemoglobin [168], haptoglobin [169], hemopexin [170], and lipocalin (Lcn2) [171]. The hypoferric inflammatory response in fish is mediated by the stimulation of the proinflammatory cytokine interleukin-6 (*il-6*) [172], which increases the synthesis and secretion of hepcidin (*hamp*) [173]. Increased hepcidin levels have an inhibitory effect on the expression of ferroportin (*fpn1*), an iron exporter that plays an important role in iron homeostasis [173,174]. As a result of decreased *fpn1* expression, the iron release is blocked, and iron uptake is decreased [175,176,177]. Overall, the iron in tissues is reduced to such a low concentration that the pathogen cannot replicate and cause disease [75]. In other words, iron limitation reduces bacterial growth to a level that enables the fish immune system to eliminate the infection [178].

Hepcidin is an antimicrobial peptide with iron regulatory properties [174]. Two functionally distinct hepcidin types have been described in teleost fish: type 1 hepcidin (*hamp 1*) is the iron metabolism regulator, and type 2 hepcidin (*hamp 2*) presents an antimicrobial role [179]. Significantly increased expression of hepcidin was observed in hybrid striped bass liver and Atlantic salmon head kidney during the early stages of infection with *S. iniae* [180] and *R. salmoninarum* [162,163], respectively. During an evaluation of the antimicrobial potential of European sea bass hepcidins, elevated expression levels of *hamp 1* and *hamp 2* were observed in response to *S. parauberis* and *L. garviae* infection [181]. Here, *hamp 1* showed no antibacterial activity, similar to what was reported for Japanese flounder hepcidins against *L. garviae* and *S. iniae* infection [181,182]. However, *hamp 2* exhibited significantly stronger antibacterial activity, especially against Gram-positive bacteria, compared to Gram-negative bacteria. Future studies will be required to unravel the prophylactic use of fish-derived hepcidins to control Gram-positive bacterial infections and to better understand the insights into their antimicrobial properties.

**Table 2 biology-11-01316-t002:** Recent studies on host (fish) response to marine Gram-positive bacterial pathogens.

Host (Tissue/Cell Type)	Pathogen	Method	Host Response *	Reference
Chinook salmon: Wisconsin and Green River stocks (Kidney)	*R. salmoninarum* ATCC 33209	qPCR	↑ interferon response in both stocks (*ifnγ*, *mx1*) ↑ *iNOS* expression and ↑ prevalence of membranous glomerulopathy in lower surviving stock than higher surviving stock ↑ iron-binding protein response (*transferrin*) in higher surviving stock than lower surviving stock	[183]
Atlantic salmon (Kidney cell line)	*R. salmoninarum*: H-2 and DSM20767 with high and low siderophore production ability, respectively	qPCR	↑ pro-inflammatory cytokines (*il1β*, *tnfα*), Gram-positive pattern recognition receptor (*TLR*), and interferon (*ifnγ*) Reduced expression of *tnfα* and *TLR1* at 24 hpi Strain (H-2) grown under iron-limited conditions induced significantly higher immune response in host cells than DSM20767 and bacteria grown under normal conditions.	[140]
Atlantic salmon (Head kidney)	Formalin-killed *R. salmoninarum* ATCC 33209	Transcriptomics (44K microarray) and qPCR	↑ pathogen recognition receptors (*tlr5*, *clec12b*) ↑ immunoregulatory receptors (*tnfrsf6b*, *tnfrsf11b*) ↑ antimicrobial effectors (*hamp*) ↑ interferon-induced response (*ch25ha*) ↑ chemokine (*ccl13*) and ↓ chemokine receptor (*cxcr1*)	[162]
Lumpfish (*Cyclopterus lumpus*) (Head kidney)	*R. salmoninarum* ATCC 33209	qPCR	Early stage (28 dpi): immunosuppressive infection ↑ pro-inflammatory cytokines (*il1β*, *il8a*, *il8b*), anti-inflammatory cytokine (*il10*), pattern recognition (*tlr5a*), iron regulation (*hamp*), and acute phase reactant (*saa5*) related genes ↑ interferon-induced response (*ifnγ*, *mxa*, *mxb*, *mxc*, *rsad2*, *stat1*) ↓ *tnfα* and cell-mediated adaptive-immunity-related genes (*cd4a*, *cd4b*, *cd8α*, *cd74*) Chronic stage (98 dpi): cell-mediated adaptive immunity ↑ *ifnγ* and *cd74*	[184]
Japanese flounder vaccinated with *sagH* DNA vaccine (Spleen and blood)	*S. iniae* SF1 (Serotype I) and 29177 (Serotype II)	qPCR and ELISA	↑ innate and adaptive immune response (*il1β*, *il1*, *il6*, *il8*, *il10*, *tnfα*, *ifnγ*, *mx*, *nkef*, *tgfβ*, *MHCI* and *II*, *cd40, cd8α*) ↑ titer of specific serum antibodies	[185]
Asian seabass vaccinated with a commercial vaccine, Norvax Strep Si (Spleen and head kidney)	*S. iniae*	Transcriptomics (8 × 60K microarray) and qPCR	Effect of vaccination was early and transient in the spleen (1–7 dpv) compared to the head kidney, which showed delayed response (21 dpv) In vaccinated spleens: ↑ NFkB, chemokine, and toll-like receptor signaling ↑ genes related to proteolysis, phagocytosis, and apoptosis Rapid T-cell-mediated adaptive immune response	[186]
Atlantic salmon and rainbow trout(Mucus, serum, and macrophages)	*S. phocae* subsp. *salmonis* isolates: two from Atlantic salmon (LM-08-Sp and LM-13-Sp) and two from seal (ATCC 51973T and P23)	Comparative innate immune response analysis	↑ lysozyme activity, phagocytic and bactericidal activity, reactive oxygen species, and NO production in rainbow trout compared to the Atlantic salmon Rainbow trout was more resistant to *S. phocae* than Atlantic salmon in terms of non-specific humoral and cellular barriers.	[187]
European Seabass (Spleen, head kidney, and blood)	*M. marinum*: virulent (Eilat) and heat-killed avirulent mutant (*iipA::kan*) strains	qPCR, ELISA	↑ specific immunoglobulin (IgM) response (1 and 2 mpc) ↑ *tnfα* in spleen at 1 mpc and return to basal levels in spleen and head kidney at 2 mpc High survival (75%), strong immune response and moderate tissue damage in avirulent mutant strain.	[188]
Amur sturgeon (*Acipenser schrenckii*) (Liver)	*M. marinum* ASCy-1.0	*De novo* transcriptome analysis (Illumina RNA seq) and qPCR	Total differentially expressed contigs (DEC): 4043 (↑ 2479, ↓1564) 78 DEC—innate immune response (*iNos2*, *saa*), phagocytosis, antigen processing and presentation (*mhc1*), chemotaxis (*ccl19*), and leucocyte regulation (*il8*) Strong leptin expression—Th1 immunity Immune pathways: TNF signaling and Toll-like receptor signaling	[189]
Amberjack vaccinated with formalin-killed *N. seriolae cells* + mixture of six recombinant amberjack IL-12 (rIL-12) as adjuvant (Head kidney and spleen leucocytes)	Formalin-killed *N. seriolae* 024013 strain	qPCR	↑ Th1-specific transcriptional factors (*ifnγ* and *T-bet*) ↓ Th2-related genes (*il10* and *GATA-3*) ↓ primary and secondary humoral immune response rIL-12 proved to be a CMI-inducible adjuvant that produced Th1 immunity cells with antigen memory	[190]
Largemouth bass (Spleen)	*N. seriolae*	*de novo* transcriptome analysis (RNA seq using Illumina hiseq) and qPCR	↑ 1384 genes, ↓1542 genes ↑ pro-inflammatory cytokines and signal-transduction-related genes (*il1β*, *il8*, *tnfα*, TNF receptors, CXC chemokines, *tgfβ*) Antibacterial mechanism at early-stage infection (24 hpi) involved cytokine–cytokine receptor interactions Immune pathway: JAK-STAT signaling	[155]
Grey mullet (Head kidney and spleen)	*L. garviae*	*De novo* transcriptome analysis (RNA seq using Illumina hiseq) and qPCR	Spleen: ↑ 3598 genes, ↓ 3682 genes (Total: 7280) Head kidney: ↑ 4211 genes, ↓ 2981 genes (Total: 7192) ↑ Pro-inflammatory cytokines, Fc receptor, and *Ig*, *il10*, *mhc-I*, *mhc-II*, *cd4*, and *cd8* ↓ *il8* and *tnfα* Immune pathways: complement and coagulation cascade, TLR signaling, antigen processing and presentation	[154]

* This column includes host immune response/activity and or immune signaling pathways differentially expressed after infection. ↑ upregulation or increase based on the response or process, ↓ downregulation or decrease based on the response or process, hpi: hours post-infection; dpv: days post-vaccination; mpc: months post-challenge.

Transferrin is one of the serum proteins capable of binding and transporting iron and creating an environment where low levels of iron restrict the growth of pathogens [191]. The biological functions of transferrin have been linked with resistance to infectious diseases [192,193,194]. Metzger and co-workers (2010) observed that the expression of the transferrin-encoding gene was upregulated up to 71 days post-infection (dpi) in two chinook salmon stocks following *R. salmoninarum* i.p. infection [183]. Interestingly, transferrin expression significantly differed between populations, where the *R. salmoninarum*-resistant salmon population showed higher transferrin expression than the susceptible population. Moreover, differential resistance among the three transferrin genotypes of coho salmon (*O. kisutch*) was observed after injection with *R. salmoninarum* [193]. On the other hand, Stafford and Belosevic (2002) demonstrated a unique role of transferrin as a mediator of fish macrophage activation in combination with the TLR system [195,196]. In this study, adding exogenous transferrin to goldfish macrophages activated by *Mycobacterium chelonei* significantly increased their nitric oxide (NO) production [195]. The role of transferrin in controlling intracellular bacterial pathogens (i.e., *R. salmoninarum* and *M. chelonei*) has not been explored and could reveal disease resistance mechanisms in fish and improve brood stock selection based on transferrin allelic variation.

The second layer of iron nutritional immunity involves Lcn2, which binds to bacterial siderophores and sequesters ferric siderophore complexes away from bacterial siderophore receptors [75,197]. A recent study found the first evidence of a functional teleost Lcn2 with antimicrobial properties in triploid crucian carp [171]. Here, Lcn2 enhanced the bactericidal activity, triggered immune defense, and increased fish resistance against Gram-negative *Aeromonas hydrophila* infection. Developing research to understand the immune effects of Lcn2 and Lcn2-mediated resistance against marine Gram-positive pathogens might be useful.

### 3.3. Innate and Adaptive (Humoral and Cell-Mediated) Immunity

The first protective barrier against infection is the fish mucus, which has bactericidal properties. It is also the first interaction site between skin epithelial cells and the pathogen [64]. Lysozyme is one of the components that helps the fish mucus have an antibacterial effect. For instance, it has been documented that the lysozyme activity in the mucus of rainbow trout controlled the growth of *S. phocae* [187]. In addition, higher levels of skin mucus in marine fish and increases in the cholesterol in the fish cell membrane aided in resisting pathogen invasion [198,199,200].

In response to *R. salmoninarum*, rainbow trout macrophages activated inflammatory responses (upregulation of *il1b*, *cox2*, *mhcII*, *iNOS*, *cxcr4*, *ccr7*) at 2 hrs post-infection [201]. TNF-α, apart from its role in regulating inflammation, is associated with the pathogenesis of chronic infections in fish [172]. *R. salmoninarum* survived initial contact with macrophages by avoiding/interfering with the TNF-α-dependent killing pathways of fish [201]. Moreover, the chronic stimulation of TNF-α, which is implicated by p57, could assist a chronic inflammatory pathology (granulomas). IFN-γ is a Th1 cytokine associated with adaptive immunity [172]. Interferon systems play a role in priming and regulating the adaptive immune response against intracellular mycobacteria [202]. Interferon and interferon-induced effectors (MX1, MX2, and MX3) are associated with the inflammatory response in fish [203,204,205]. Thus, the expression of *ifnγ* and *mx1-3* upon *R. salmoninarum* infection in chinook salmon and rainbow trout may be linked to the priming of adaptive immunity [183,201,202]. In addition, the early upregulation of interferon-induced effectors in response to a *R. salmoninarum* strain with reduced p57 suggests these genes as possible immune indicators in vaccine design [206,207].

The teleost adaptive immune system is subdivided into humoral immunity, which involves antibodies to neutralize pathogens in body fluids, and cell-mediated immunity, which kills and eliminates pathogen-infected cells [208]. Extracellular pathogens evoke humoral immune responses, while intracellular pathogens evoke both humoral and cell-mediated immune responses. Although salmonids mount a humoral response against *R. salmoninarum*, there is no clear correlation between this antibody response and protective immunity [7,111]. Moreover, the humoral response is counterproductive, as it is linked to an exacerbated BKD pathology (i.e., antigen-antibody complex deposition in glomeruli) [7,209,210]. Antigen-antibody immune complexes formed during infection might weaken the effective antibody response by adsorbing the circulating antibodies before they bind to p57 and block its activity related to immune suppression [7,211].

Few studies reported the cell-mediated immune response of *R. salmoninarum* [201,212,213,214]. A p57-induced chronic reduction in MHC II expression might consequently skew the T-cell responses toward an MHC-I-dependent cell-mediated response [201]. Khalil et al. (2020) presented a complete picture of the Atlantic salmon innate and adaptive immune response to live *R. salmoninarum* by using a 44 K salmonid microarray platform in a transcriptome profiling study [163]. For instance, *R. salmoninarum* differentially regulated the Atlantic salmon adaptive immune responses, including B- or T-cell differentiation, function, and antigen presentation. Moreover, *R. salmoninarum* infection levels have an impact on the JAK-STAT signaling pathway during host–pathogen interactions [163].

Sakai et al. (1989) demonstrated the protective immune response against streptococcal infection in rainbow trout immunized with β-hemolytic *Streptococcus* spp. bacterin [215]. Here, the serum of fish immunized with i.p. injected streptococcal bacterin showed no enhanced bactericidal activity but had agglutinating antibodies. However, these specific antibodies were not associated with protective immunity. Interestingly, increase in the phagocytic activity of kidney leucocytes observed in the vaccinated fish could be aided in the rapid bacterial clearance from the spleen, liver, kidney, and blood 72 h post-challenge, suggesting that cellular immunity plays a major role in the rainbow trout defense against *Streptococcus* spp.

The cell-mediated immunity involving CD8+ T cells is effective in killing and eliminating intracellular Gram-positive pathogens (*R. salmoninarum* and *Mycobacterium* spp) in fish [208,216], and protective cell-mediated immunity could be achieved by inducing these cells [208]. For instance, CD8+ T cells are activated into cytotoxic T lymphocytes upon binding to MHC-I molecules that express processed antigens from intracellular pathogens. T lymphocytes secrete cytotoxic granules. The perforin and granzyme contents of these granules induce the apoptosis of infected cells. Concurrently, cytokine signatures, such as TNF-α and IFN-γ that skew CD4+ cells towards Th1 differentiation, would help in the priming of CD8+ cells as “immune-adjuvants” for producing protective immunity.

### 3.4. Fish Resistance/Tolerance/Susceptibility to Marine Gram-Positive Bacteria

Resistance is the ability to limit a pathogen in terms of its replication or spread. On the other hand, tolerant fish would show less pathology when comparing high- and low-tolerance fish populations with equivalent pathogen burdens [217]. Metzger et al. (2010) demonstrated the ‘resistance’ and ‘tolerance’ in two Chinook salmon stocks (higher-surviving WI stock and lower-surviving green river stock) following an *R. salmoninarum* challenge [183]. The WI stock showed lower bacterial loads than the green river stock at 28 dpi, which implied the resistance of higher-survival stock. Conversely, the green river stock exhibited higher mortality levels than the WI stock by 44 dpi, when both stocks had similar levels of bacterial load, which explained the tolerance of higher-survival stock. Thus, the authors pointed out that the enhanced tolerance of chinook salmon against *R. salmoninarum* could benefit the fitness of both the host and the pathogen in the dynamic interaction. Sako (1992) observed an acquired immune resistance in yellowtail (*Seriola quinqueradita*) when the fish that had recovered from experimental infection with *S. iniae* were reinfected [218]. Here, the bacterial loads in the spleen, kidney, and blood showed rapid decreases, whereas no bacterial proliferation was observed in the brain.

As water temperature affects both the rate of bacterial multiplication and the fish immune response, the rapid shifts in marine water temperature could alter host-pathogen interactions and reduce host resistance [219]. For instance, fish become susceptible to streptococcal infections during summers with high temperatures [220]. Lower water temperatures (8 °C) contributed to the disease progression and transmission potential in chinook salmon infected with *R. salmoninarum* [219]. Moreover, inhibited cell-mediated immunity and a higher risk of death during the late stage of infection were observed in Atlantic salmon pre-molt survivors upon *R. salmoninarum* infection in low water temperatures (11 °C) [221].

## 4. Concluding Remarks

Taken together, this article provides an overview of the host-centric and pathogen-centric approaches at the host–pathogen interface between economically important marine fish and Gram-positive pathogens. Marine Gram-positive pathogens developed a unique set of machinery/strategies to interact with their exclusive fish host cells and modulate the complex molecular and cellular networks of these cells to allow bacterial proliferation and spread while counteracting fish defenses. The pathogenicity of marine Gram-positive pathogens strongly depends on the host it is trying to infect. For instance, *R. salmoninarum* has primarily adapted to infect and persist in salmonids [7].

Knowledge of how the host and pathogen interact is crucial for a true understanding of disease and is key to developing control or prevention strategies. Studies in marine Gram-positive pathogens that have been conducted so far focused mostly on bacterial virulence and fish immune responses. A few studies considered how environmental stressors (i.e., temperature or hypoxia) affect host-pathogen interactions and alter disease progression [124,219]. Dynamic host-pathogen interactions between marine Gram-positive pathogens and fish hosts are complex. Exploring how host-pathogen-environment interactions mediate disease outcomes in marine fish populations with the help of integrative omics research will add another layer of complexity. Purcell et al. (2015) suggested that under a warming climate, *R. salmoninarum* may pose a lesser risk to chinook salmon since low temperatures (8 °C) favor its infection compared to higher temperatures (12 °C and 15 °C) [219]. It will be useful to conduct research to determine which Gram-positive bacterial pathogens may pose a threat to cultured fish due to climate change.

As prophylaxis design engages in either disabling the bacterial virulence or boosting the host system, vaccines and immunostimulants are effective and sustainable in aquaculture [126,222,223]. Most licensed fish vaccines against streptococcal and lactococcal infections are traditional inactivated microorganisms [224]. A live vaccine contains non-pathogenic *Arthrobacter davidanieli,* which is closely phylogenetically related to *R. salmoninarum,* is commercially licensed for BKD control, and elicits cross-immunity [225]. However, the protective immunity of this vaccine is experimentally and intellectually questionable for protecting a wide range of salmonids [207]. The protective effect of DNA and subunit vaccines in fish has been demonstrated against *S. iniae* infection [185,226,227,228]. A rational fish vaccine design using alternative technologies beyond just bacterins, including recombinant live-attenuated or RNA vaccines, is essential [224]. These technologies have yet to be reported to prevent marine Gram-positive infections in fish. In contrast, there are several experimental reports of such vaccine designs against Gram-negative fish pathogens. For instance, the LcrV protein (V antigen) is an essential virulent factor of *Yersinia pestis*, a Gram-negative pathogen and the causative agent of the bubonic plague [229]. Variants of the V antigen lacking the immune suppressor region induced protective immunity in mice [230,231,232]. p57 and p22 have been reported as immunosuppressive proteins contributing to *R. salmoninarum* virulence. Thus, identifying or designing the variants of p57 and p22 with reduced immunomodulatory properties and using them as immune protective antigens will be an attractive vaccine design for BKD control in mariculture. Although the area of functional genomics of fish pathogenic bacteria has been slowly progressing, most of the economically important marine Gram-positive bacterial genomes were sequenced [233]. These available genomes open exciting opportunities in the search for universal vaccine candidates across the fish pathogens and shed light on using reverse vaccinology approaches. For example, a recent study used a reverse vaccinology pipeline to identify a set of antigens that could be used to develop a polyvalent vaccine against Gram-negative bacterial infections that impact Atlantic salmon and lumpfish aquaculture [234]. Vaccines against marine Gram-positive bacterial infections could be developed using similar techniques. In addition to vaccines, immunostimulants could be used as fish non-specific immune defense enhancers to improve fish resistance to disease. Improved phagocytic activity was observed in rainbow trout treated with the fermented chicken egg product EF2013 during streptococcal and renibacterial infections [235,236]. Research on the use of immunostimulants to alter host-pathogen interactions and improve fish immunocompetency against marine Gram-positive bacteria will be beneficial.

As aquaculture continues to grow globally, applying bioinformatics to expand our knowledge on the host-pathogen-environment interactions of marine Gram-positive bacteria will be valuable in solving emerging fish health issues.

## Figures and Tables

**Figure 1 biology-11-01316-f001:**
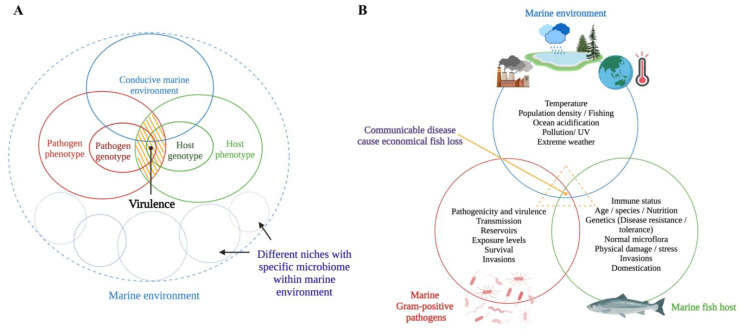
Schematic representations of (**A**) how virulence is modulated in a dynamic host–pathogen interaction [21] (the yellow shaded area expresses virulence as an outcome of a dynamic host-pathogen interaction in a conducive marine environment) and (**B**) how the disease process occurs as a result of complex host-pathogen-environment interactions. This figure was generated by the authors using BioRender (https://biorender.com/) (accessed on 1 August 2022).

**Figure 2 biology-11-01316-f002:**
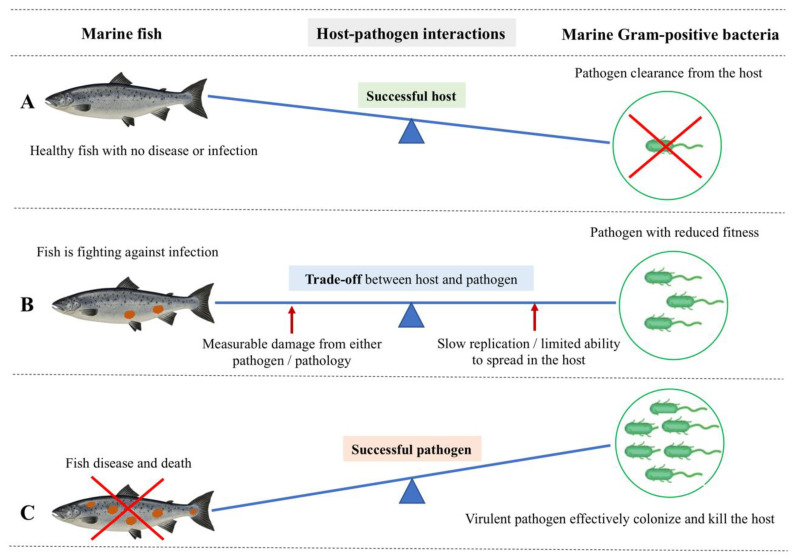
Host-centric and pathogen-centric views of dynamic host-pathogen interactions between marine Gram-positive bacteria and a marine fish host. (**A**) Successful host: Fish show no disease or infection because of successful pathogen clearance by the host-induced pathogen-specific immune response. Therefore, fish stay healthy, with effective immunological and physiological homeostasis upon an infection event. (**B**) The trade-off between host and pathogen: It is the actual tug-of-war scenario between the fish host and pathogen. Here, the fish is still alive but fighting against the infection with an initiated specific immune response. In this trade-off, the pathogen compromises its fitness, thereby slowly replicating without alerting the fish immune response and exploiting the fish host without killing it. On the other hand, the fish is measurably damaged as a result of either the pathogen’s action or a host-specific pathology. There is a scenario where increased acquisition and allocation of nutrients occur at the host/pathogen nutritional interface as opposed to a trade-off. Here, the host withdraws an essential nutrient supply to suppress pathogen proliferation while increasingly allocating the nutrients to fuel immune proliferation. The pathogen, on the other hand, gradually acquires host nutrients to fuel its own replication and survival. (**C**) Successful pathogen: A virulent pathogen effectively colonizes using host resources and killing the host by successfully escaping the barriers of the fish host’s innate and adaptive immunity, leaving and infecting a new host. Therefore, the fish shows severe disease and death.

**Figure 4 biology-11-01316-f004:**
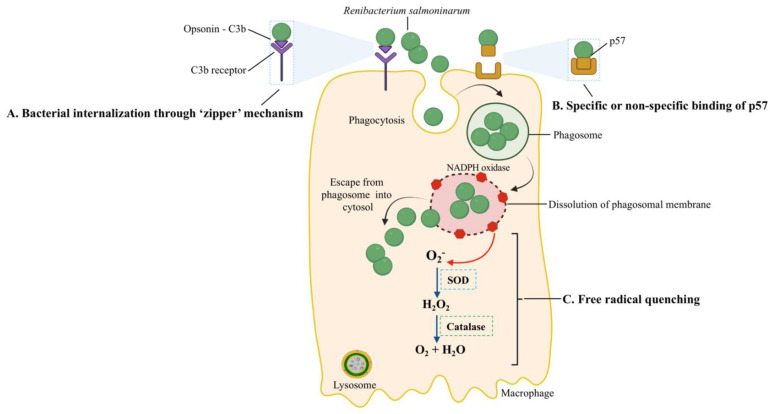
Host-pathogen interactions during the intracellular entry, replication, and survival of *R. salmoninarum*. *R. salmoninarum*’s entry into macrophages is facilitated in two ways: (**A**) Bacterial internalization through a ‘zipper’ mechanism. Here, opsonin C3b binds to the bacterial surface, followed by ligation to C3b-receptor-bearing fish phagocytes and the intracellular invasion of bacteria into host cells. (**B**) Specific or non-specific binding of p57 to host cells. The hydrophobic surface protein p57, which resembles adhesion protein, may allow bacterial adherence to host cell receptors through specific or non-specific binding. Phagocytized *R. salmoninarum* escapes from the phagosome into the cytosol by budding out of the phagosome or phagosomal membrane lysis. (**C**) *R. salmoninarum* evades oxidative stress from reactive oxygen species by ‘free radical quenching.’ Here, NADPH oxidase generates ROS. Superoxide (O_2_^−^) is converted into H_2_O_2_ and then O_2_ and H_2_O by superoxide dismutase (SOD) and catalase, respectively. This original illustration was generated by the authors using BioRender (https://biorender.com/) (accessed on 17 September 2021).

**Figure 5 biology-11-01316-f005:**
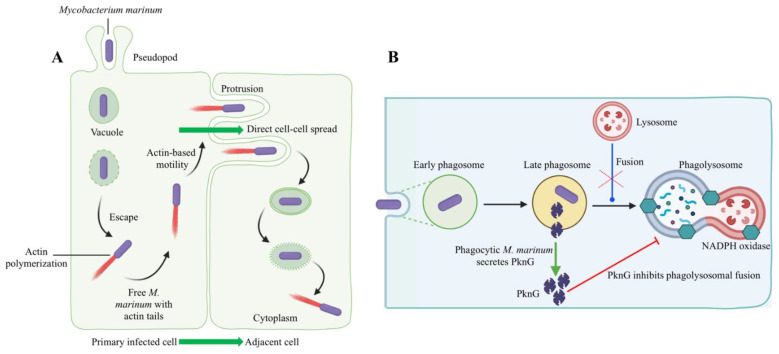
Host–pathogen interactions during intracellular evasion strategies of *M. marinum. M. marinum* has been shown to use two methods to evade phagocytic killing: (**A**) Escape from phagosomes into the cytosol and cell-to-cell spread via actin-based motility [123]. The initial uptake of *M. marinum* into phagocytic vacuoles is followed by an escape from vacuoles into the cytoplasm. According to Stamm et al. (2003), *M. marinum* polymerized actin through the recruitment of host cell cytoskeletal factors in the cytoplasm [112]. Thus, free *M. marinum* with actin tails was observed in the host cell cytoplasm. The acquisition of actin-based motility allows *M. marinum* to spread cell to cell directly from primarily infected cells to adjacent cells without leaving the cytoplasm. (**B**) *M. marinum* blocks phagolysosomal fusion (late phagosome fused with phagocytic lysosome), thus resisting the discharge of lysosomal contents into the phagosome. Here, protein kinase G (PknG) secretion through the SecA2 pathway is initiated by phagocytic *M. marinum*. Secreted PknG directly inhibits the fusion of late phagosomes with lysosomes. This original illustration was generated by the authors using BioRender (https://biorender.com/) (accessed on 16 November 2021).

## Data Availability

Not applicable.

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
