# Peer review of "Host–Pathogen Interactions of Marine Gram-Positive Bacteria"

_biology, 2022, doi:10.3390/biology11091316_

Round 1

Reviewer 1 Report

This review article written for the Biology journal is a comprehensive discussion on host-pathogen interactions in fish with Gram-positive bacteria. In particular, authors provide a general insight into host-pathogen interactions and supplement the reader’s understanding with well-drawn – Figures 1 and 2. Thereafter, the authors elaborate on the two sides of this interaction separately, i.e. pathogen-centric approaches and host-centric approaches. Both of these sections include multiple examples of gram-positive bacteria and are supplemented with clearly drawn schematics and tables, which would be helpful for the readers of this article.

While well written for the most part, this reviewer came across a few questions and would like the authors to elaborate on those:

1.     What is so interesting about gram-positive bacteria, that those became the focus of their article? While the links to pathogenicity, aquaculture and climate change have been made in this article, what specifically distinguishes them from gram-negative bacteria? Perhaps a table or a schematic to compare and contrast the key host-pathogen mechanisms, with gram-positive and gram negative bacteria would help the readers to see the rationale behind going in-depth with gram-positive bacteria specifically.

2.     In line#77, authors are describing virulence as an outcome of a specific host-pathogen interaction, not a fixed microbial property. This is confusing in light of the definition of the term “pathogenicity”. It would be nice, if authors could elaborate on the differences between virulence and pathogenicity.

3.     Figure 2 is a good figure to capture various outcomes associated with host-pathogen interaction. This reviewer is curious if authors have explored yet another scenario of increased acquisition and allocation of nutrients for both – the host and the pathogen (as opposed to trade-off)? Could the authors potentially elaborate on that?

4.     Lines 144-153 are misplaced in the text. What is table 57 referring to?

5.     While the article flows well for the most part, certain terms, e.g. pathogen, virulence, zipper and trigger mechanisms, could be described prior to their first introduction. Additionally, an in-text citation for the figures at appropriate points in the text would enhance readability.

6.     Authors indicate that the schematics were drawn using biorender. Were they drawn by the authors? If so, then perhaps they could indicate that as well, in the figure legend. Also, the journal and the authors may want to look into the need for copyright with figure 5, as it was adapted and modified from Weddle and Agaisse, 2018.

7.     The host-centric approaches describe TLRs in detail. How about other Pattern Recognition Receptors and their involvement in recognizing gram-positive bacteria? The reason for going in-depth with TLRs only is not mentioned.

8.     There are minor typos here and there, and proofreading should help take care of those. For example; lines 164-165 are in past tense, while rest of the description is not. In line 642 – it should be ‘these’ cells.

Overall, this is a well-written, comprehensive review on a niche topic of host-pathogen interactions involving marine gram-positive bacteria.

Author Response

Response to Reviewer 1

This review article written for the Biology journal is a comprehensive discussion on host-pathogen interactions in fish with Gram-positive bacteria. In particular, authors provide a general insight into host-pathogen interactions and supplement the reader’s understanding with well-drawn - Figures 1 and 2. Thereafter, the authors elaborate on the two sides of this interaction separately, i.e., pathogen-centric approaches and host-centric approaches. Both of these sections include multiple examples of gram-positive bacteria and are supplemented with clearly drawn schematics and tables, which would be helpful for the readers of this article.

While well written for the most part, this reviewer came across a few questions and would like the authors to elaborate on those:

We would like to thank and appreciate reviewer 1 for their suggestions/comments/questions on our review. We did our best to address the questions raised by the reviewer 1 below, and the elaborations on those questions were incorporated to the revised manuscript. Please check the track changes in the revised manuscript and see the individual responses below.

  1. What is so interesting about gram-positive bacteria, that those became the focus of their article? While the links to pathogenicity, aquaculture and climate change have been made in this article, what specifically distinguishes them from gram-negative bacteria? Perhaps a table or a schematic to compare and contrast the key host-pathogen mechanisms, with gram-positive and gram negative bacteria would help the readers to see the rationale behind going in-depth with gram-positive bacteria specifically.

Author’s response: We would like to thank and appreciate reviewer-1 for this very thoughtful comment. Comparative aspects of Gram-negative vs. Gram-positive are mentioned in the first paragraph.  Including cell wall composition is the key factor distinguishing Gram-positive (i.e., thick peptidoglycan) from Gram-negative bacteria (i.e., thin peptidoglycan and LPS). There are several specific aspects of marine Gram-positive pathogens that we highlight in this review. We agree with reviewer-1 that presenting the key differences of host-pathogen mechanisms between Gram-positive and Gram-negative would be great. However, it will also widen the article’s scope and take the review in a different direction. This review is already having 2 tables and 5 figures. Thus, we attempt to present a graphical abstract that summarises key host-pathogen mechanisms of Gram-positive bacteria. We anticipate that this graphical abstract will satisfy reviewer-1.

  1. In line#77, authors are describing virulence as an outcome of a specific host-pathogen interaction, not a fixed microbial property. This is confusing in light of the definition of the term “pathogenicity”. It would be nice if authors could elaborate on the differences between virulence and pathogenicity.

 Author’s response: Methot and Alison argued virulence as an outcome of a specific host-pathogen interaction, not a fixed microbial or host property (Methot and Alison, 2014). This could be exemplified by pathogenicity of avirulent microbes in immunocompromised hosts and the lack of pathogenicity of virulent pathogens in immune host (Casadevall and Pirofski, 2001). Also, if virulence is a fixed microbial property, pathogen should remain virulent regardless of host or environment. But in reality, virulence varies when the host or environment changes (Figure 1A). We agree with reviewer-3 that this statement is controversial when considering the classic definitions of virulence and pathogenicity. However, since this review is a comprehensive synopsis of host-pathogen interactions, we have included both reliable and controversial facts under this concept. We attempt to elaborate on the differences between virulence and pathogenicity as per the suggestion of reviewer-1 in lines numbers 72-74 and 80-82 in the revised manuscript.

  1. Figure 2 is a good figure to capture various outcomes associated with host-pathogen interaction. This reviewer is curious if the authors have explored yet another scenario of increased acquisition and allocation of nutrients for both - the host and the pathogen (as opposed to trade-off)? Could the authors potentially elaborate on that?

Author’s response: We appreciate reviewer-1 for this thoughtful question. We agree with reviewer-1 about the outcome related to host-pathogen nutritional interface. Competition for nutrients is critical for the survival of both the host and the pathogen. Nutrient bioavailability at infectious interface exerts a selective pressure, which could shape the nutrient sequestration strategies of both the host and the pathogen. For example, iron-acquiring strategies have evolved in pathogens, and the host has evolved its own strategies for iron sequestration and allocation (Prentice et al., 2006; Murdoch and Skaar, 2022). Host immune proliferation and pathogen replication require within-host resources. Therefore, increased nutrients acquisition and/or allocation can be observed as an outcome at host/pathogen-nutritional interface, for instance, nutrient acquisition to fuel pathogen replication on the pathogen side and nutrient allocation to fuel immune proliferation on the host side (Cressler et al., 2014). Since the main focus of the article is about the host-pathogen interactions that results in economically important fish diseases, figure 2 captures the most frequent disease-related outcomes. However, we covered some information related to nutrient acquisition and allocation for both the pathogen and the host in the sections; proliferation and survival inside the host under pathogen-centric approaches, and nutritional immunity under the host-centric approaches. In addition, we attempt to address the scenario suggested by the reviewer-1 in the figure 2 caption (lines 101-106) in the revised manuscript.

  1. Lines 144-153 are misplaced in the text. What is table 57 referring to?

Author’s response: Thank you for pointing out this issue. We believe it’s a typesetting error, which misplaced the part of the paragraph as table caption 57. Table caption is replaced with the missing original text “The major soluble antigen (msa) p57” in the revised manuscript.

  1. While the article flows well for the most part, certain terms, e.g., pathogen, virulence, zipper and trigger mechanisms, could be described prior to their first introduction. Additionally, an in-text citation for the figures at appropriate points in the text would enhance readability.

Author’s response: We thank reviewer-1 for this nice suggestion. We did our best to introduce the terms before starting to explain the literature. In the introduction, the terms pathogen and virulence have been described (lines 71-74). Lines 267-271 and 298-300, respectively, defined the terms trigger and zipper. In-text citations for the figures were added at pertinent points in the main text.

  1. Authors indicate that the schematics were drawn using BioRender. Were they drawn by the authors? If so, then perhaps they could indicate that as well, in the figure legend. Also, the journal and the authors may want to look into the need for copyright with figure 5, as it was adapted and modified from Weddle and Agaisse, 2018.

Author’s response: Yes, we drew all the original schematics for the article. A statement mentioning that “the authors generated the figures using BioRender” has been added to respective figure captions in the revised manuscript as per the suggestion of reviewer-1. The phrase “adapted and modified from Weddle and Agaisse, 2018” was added to figure caption 5A to indicate where the fundamental concept or inspiration for making this particular figure came from. We created figure-5 exclusively for this article to graphically represent M. marinum direct cell-cell entry based on literature, and basic concept or idea we adapted from Weddle and Agaisse, 2018. We remove that statement and just only cite the article reference in the figure caption to avoid confusions. We would like to thank reviewer for pointing out this matter. 

  1. The host-centric approaches describe TLRs in detail. How about other Pattern Recognition Receptors (PRR) and their involvement in recognizing gram-positive bacteria? The reason for going in-depth with TLRs only is not mentioned.

Author’s response: Reported PRR in teleost fish are TLR (Toll-like receptors), NLR (NOD-like receptors), CLR (C-type lectin receptors) and PGRP (peptidoglycan recognition proteins) (Boltaña et al., 2011). Among these PRR, the signalling PRRs, TLRs and NLRs, play a significant role in fish innate immunity. The first class of PRRs to be extensively studied and characterized in fish were TLRs (Sahoo, 2020). Additionally, substantial research has been done on TLR recognition of both Gram-positive and Gram-negative pathogens (Boltaña et al., 2011; Palti, 2011; Sahoo, 2020). Research on other PRRs involvement in Gram-positive bacterial recognition in marine fish is lacking and has yet to be reported. Thus, we choose to go in-depth with TLRs only after considering the following factors: i. Fish innate immunity uses TLR as an important signalling PRR; ii. Sufficient literature is available on TLR-based Gram-positive bacterial recognition. We explained why we are just focused on TLRs in lines 501-504, as reviewer-1 suggested.

“In fish, TLRs (Toll-like receptors), NLRs (NOD-like receptors), CLRs (C-type lectin receptors) and PGRP (peptidoglycan recognition proteins) are the 4 main types of Pattern Recognition Receptors (PRRs) [149]. Only TLRs are the subject of this section because they are well-described fish innate immunity PRRs detecting Gram-positive pathogen-associated molecular patterns (PAMPs) [150]. Research on other PRRs involvement in Gram-positive bacterial recognition in marine fish has yet to be reported.”

  1. There are minor typos here and there, and proofreading should help take care of those. For example, lines 164-165 are in past tense, while rest of the description is not. In line 642 – it should be ‘these’ cells.

Author’s response: We appreciate reviewer 3 for pointing out the typos and recommending proofreading. The article was proofread, and the revised manuscript contains the corrections made.  

Overall, this is a well-written, comprehensive review on a niche topic of host-pathogen interactions involving marine gram-positive bacteria.

We are glad about this opinion. Thank you.

References

Methot and Alison, 2014 - DOI: 10.4161/21505594.2014.960726

Casadevall and Pirofski, 2001 - DOI: 10.1086/322044

Prentice et al., 2006 - DOI: 10.1093/jn/137.5.1334

Murdoch and Skaar, 2022 - DOI: 10.1038/s41579-022-00745-6

Cressler et al., 2014 – DOI: 10.1111/ele.12229

Boltaña et al., 2011 – DOI: 10.1016/j.dci.2011.02.010

Palti, 2011 – DOI: 10.1016/j.dci.2011.03.006

Sahoo, 2020 – DOI: 10.1016/j.ijbiomac.2020.07.293

Reviewer 2 Report

The manuscript is very well written, I have only a few minor suggestions/comments:

Line 144: Part of the text is missing and this paragraph was placed as a table caption 

lines 276-279: it is not clear why M. marinum esx5 mutants were slightly attenuated in the zebrafish embryos but hypervirulent in adult zebrafish. I suggest explaining better this statement.

Table 2: the scientific name of lumpfish is missing. Check if any other scientific name is missing. Some were placed in table 1.

Very good figures, they help explaining the text.

Author Response

Response to Reviewer 2

The manuscript is very well written, I have only a few minor suggestions/comments:

We would like to thank and appreciate your suggestions/comments on our review. The manuscript has been revised based on your comments. Please check the track changes in the revised manuscript and see the individual responses below.

  1. Line 144: Part of the text is missing, and this paragraph was placed as a table caption 

Author’s response: Thank you for pointing out this issue. We believe it’s a typesetting error, which makes the paragraph as table caption. Table caption is replaced with the missing text in the revised manuscript.

  1. Lines 276-279: it is not clear why M. marinum esx5 mutants were slightly attenuated in the zebrafish embryos but hypervirulent in adult zebrafish. I suggest explaining better this statement.

Author’s response: Weerdenburg et al. (2012) observed the hypervirulence in adult zebrafish characterized by higher ESX-5 mutant bacterial loads and rapid onset of granuloma formation. The difference in virulence between the embryo and adult model is not seemed to be mediated by the adaptive immune response. Because the hyper virulence phenotype (i.e., outgrowth of the ESX-5 mutant over the wild type) was also observed in the rag-deficient zebra fish, which lack functional B and T lymphocytes. Therefore, M. marinum hypervirulence in adult zebrafish might be mediated by other factors that differ between embryonic and adult zebrafish. For instance, it may be not due to general immune response or altered extracellular environment but could be due to more local and possibly intracellular effects that result from the interplay between fish host and M. marinum. Therefore, concurrent use of embryo and adult model systems is crucial to unravelling the mycobacterial virulence mechanisms (Leeuwen et al, 2015). We agree with reviewer-2 on this and attempted to explain this statement clearly (line numbers: 281-291) in the revised manuscript.

M. marinum esx5 mutants were utilized by Abdallah et al. (2008) to demonstrate the role of M. marinum ESX-5 in triggering host cell death and modulating macrophage cytokine responses [92]. M. marinum esx5 mutants were slightly attenuated in the zebrafish embryos but hypervirulent in adult zebrafish, which was characterized by higher esx5 mutant bacterial loads and early initiation of granuloma formation in adult zebrafish. This difference in virulence between the embryo and adult zebrafish does not appear to be mediated by the adaptive immune system since the rag-deficient zebrafish, which lack functional B and T lymphocytes, also exhibited the hypervirulence phenotype. Therefore, other factors that differ between embryonic and adult zebrafish may mediate M. marinum hypervirulence in adult zebrafish. For instance, it could be caused by more local and possibly intracellular effects that result from the interplay between the fish host and M. marinum rather than a general immune response or modified extracellular environment [93].”

  1. Table 2: the scientific name of lumpfish is missing. Check if any other scientific name is missing. Some were placed in table 1.

Author’s response: The scientific names for lumpfish and amur sturgeon were added to the updated Table 2 after we examined the missing names.

Very good figures, they help explaining the text.

We are glad about this opinion. Thank you.

References

  1. Weerdenburg et al., 2012: (DOI: 10.1111/j.1462-5822.2012.01755.x)
  2. Leeuwen et al., 2015 - (DOI: 10.1101/cshperspect.a018580)